# The Dimensions of Research Attitudes Among University Faculty: A Sociodemographic Analysis in La Libertad, Peru

**DOI:** 10.3390/bs15040515

**Published:** 2025-04-12

**Authors:** Lisseth Katherine Chuquitucto Cotrina, Emma Verónica Ramos Farroñán, Marco Agustín Arbulú Ballesteros, María de los Ángeles Guzmán Valle, Julie Catherine Arbulú Castillo, Gary Christiam Farfán Chilicaus, Gladys Sandi Licapa-Redolfo, Christian David Corrales Otazú, Sarita Jessica Apaza Miranda

**Affiliations:** 1Instituto de Investigación en Ciencias y Tecnología, Universidad César Vallejo, Campus Chepén-Chiclayo, Trujillo 13001, Peru; lchuquituctoco@ucvvirtual.edu.pe (L.K.C.C.); eramosf@ucv.edu.pe (E.V.R.F.); jarbuluca26@ucvvirtual.edu.pe (J.C.A.C.); gfarfanch@ucvvirtual.edu.pe (G.C.F.C.); 2Escuela de Ingeniería de Sistemas, Universidad Tecnológica del Perú, Lima 15046, Peru; c15025@utp.edu.pe; 3Grupo de Investigación en Ecología Evolutiva, Protección de Cultivos, Remediación Ambiental, y Biotecnología (EPROBIO), Universidad Privada del Norte, Trujillo 13006, Peru; gladys.licapa@upn.edu.pe; 4Facultad de Ciencias de la Empresa, Universidad Continental, Lima 15306, Peru; ccorrales@continental.edu.pe (C.D.C.O.); sapaza@continental.edu.pe (S.J.A.M.)

**Keywords:** research attitudes, faculty development, higher education, sociodemographic factors, La Libertad Peru

## Abstract

In recent years, the academic research landscape in La Libertad, Peru, has been characterized by a limited and uneven scientific output among institutions and faculty members. Factors such as an excessive workload, insufficient resources, and inadequate training in research methodologies have hindered the development of a robust research culture. Understanding the attitudes of faculty members toward research is crucial for enhancing scientific production within the university setting. The aims of this study were (1) to comparatively analyze faculty’s attitudes toward research in La Libertad, considering their sociodemographic characteristics, and (2) to examine these attitudes across five key dimensions: personal perception, capabilities and training, evaluation, challenges, and collaboration and methodology in research. A sample of 110 university teachers from Chepén, Piura, Trujillo, and Chiclayo participated in a structured questionnaire survey. Descriptive statistics were calculated, and comparative analyses were conducted via the Mann–Whitney U test and the Kruskal–Wallis test. The results revealed significant differences in attitudes toward research on the basis of gender, age, and teaching experience across various dimensions. Personal perceptions of research varied significantly across all three sociodemographic factors. Research skills and training showed disparities by gender and teaching experience but not by age. The valuation of research differed by gender and teaching experience but not by age. The challenges in research varied by age but not by gender or experience. Collaboration and methodology differed by gender and teaching experience but not by age. This study contributes to our understanding of the research attitudes in higher education by highlighting the complex interplay of sociodemographic factors. These findings have implications for developing personalized professional development strategies and targeted interventions to enhance research skills, improve the valuation of research, and address age-specific challenges in the research process.

## 1. Introduction

Academic research is a fundamental pillar in the development and advancement of knowledge in higher education. However, teachers’ disposition toward this crucial activity can vary significantly and can be influenced by various personal, institutional, and contextual factors. In this sense, analyzing the attitudes toward research among faculty members is essential to understanding and enhancing scientific production within the university setting ([7]; [10]; [36]).

In the context of La Libertad, Peru, the problematic reality manifests in a limited and uneven scientific output from institutions and faculty members. Previous studies have indicated that factors such as workload, a lack of resources, and insufficient training in research methodologies may hinder the development of a strong research culture ([28]). Additionally, attitudes toward research may vary according to sociodemographic characteristics such as age, gender, education level, and teaching experience, underscoring the need for a detailed comparative analysis ([32]; [37]).

The justification for this study lies in its potential to inform policies and strategies that could promote a more robust research culture in La Libertad. By identifying the factors that influence attitudes toward research, specific and effective interventions can be designed for different profiles of teachers, thus optimizing institutional resources and efforts ([15]; [19]). Furthermore, this analysis contributes to the literature on professional development in the Latin American context, addressing this region’s particularities ([24]; [31]).

In this study, we adopt the concept of “attitude” as a learned and relatively stable predisposition that orients an individual’s favorable or unfavorable response to an object or situation ([37]). From a social psychology standpoint, this approach posits that attitudes stem from beliefs, affective responses, and observable behaviors.

To ground this perspective, we draw on the Theory of Planned Behavior, which holds that one’s attitude toward a behavior (in this case, research) influences one’s intention to perform it and subsequently one’s behavior itself. In the context of university teaching, this theory elucidates how personal, contextual, and institutional factors may either foster or hinder faculty engagement in research activities.

The knowledge gap that this study seeks to address centers on the lack of research examining faculty’s attitudes toward research in La Libertad in an integrated and comparative manner, considering multiple dimensions and sociodemographic factors. Although studies exist on research attitudes in other contexts, few have addressed the specific reality of this Peruvian region, and even fewer have conducted such an exhaustive comparative analysis ([12]; [20]).

This research aims to comparatively analyze faculty’s attitudes toward research in La Libertad considering their sociodemographic characteristics and five key dimensions: personal perceptions, capabilities and training, evaluation, challenges, and collaboration and methodology. This multidimensional approach allows for a deeper and more nuanced understanding of the current situation ([5]; [14]).

To achieve this general objective, this study assesses teachers’ personal perceptions of research, identify their capabilities and training, analyze the value they assign to research activities, determine the main challenges they face, and examine their attitudes toward collaboration and research methodology. All of these aspects are compared across different sociodemographic groups to identify significant patterns and variations ([8]; [25]).

Additionally, this study seeks to identify teacher profiles on the basis of their attitudes toward research and sociodemographic characteristics, allowing for the development of differentiated strategies to foster positive research attitudes among the faculty of La Libertad ([3]; [17]). This personalized approach has the potential to significantly improve the effectiveness of institutional interventions and educational policies.

The general and specific problems in this study are as follows. This study seeks to explore how university faculty’s attitudes toward research in the La Libertad region vary across five key dimensions—personal perception, skills and training, valuation, challenges and difficulties, and collaboration and methodology—when considering the sociodemographic factors of gender, age, and teaching experience and specific research questions: First, given previous evidence suggesting that gender plays a critical role in shaping professional attitudes, this study inquires about whether male and female faculty members exhibit notably different views in terms of their personal perceptions and valuation of research (H1). Second, to capture possible generational distinctions, it investigates the extent to which challenges and difficulties in conducting research differ by age group, with the premise that older faculty may confront unique barriers relative to those for younger cohorts (H2). Finally, recognizing the potential influence of accumulated teaching years, this study examines whether more experienced faculty demonstrate higher levels of collaboration and methodological engagement, thereby reflecting the deeper integration of research practices into their academic work (H3). Together, these questions provide a structured approach to identifying both the universal and subgroup-specific factors that influence faculty’s research attitudes, enabling a more nuanced understanding of how to design targeted interventions and policies that bolster scientific inquiry within diverse educational contexts.

The hypotheses proposed in this study are as follows:

**Hypothesis** **1** **(H1).**
*Gender will significantly influence attitudes toward research, particularly in the dimensions of personal perception and the valuation of research.*


**Hypothesis** **2** **(H2).**
*Age groups will differ in the dimension of their challenges and difficulties such that older faculty members will perceive different barriers compared to younger ones.*


**Hypothesis** **3** **(H3).**
*Teaching experience will significantly affect collaboration and methodology attitudes, with more experienced faculty scoring higher in collaborative approaches.*


The importance of this study is reinforced by the growing need to adapt educational practices to an increasingly digital and globalized environment. Teachers’ attitudes toward research not only influence their own scientific output but also impact the quality of teaching and training of future researchers ([1]; [18]). In this sense, understanding how sociodemographic characteristics and different dimensions of research attitudes interact can provide valuable insights for designing more effective and personalized professional development programs.

Moreover, in a post-pandemic context in which educational institutions have had to adapt quickly to new modes of teaching and learning, it is crucial to examine how these experiences have influenced teachers’ attitudes toward research and educational innovation ([22]; [33]). This study offers a unique opportunity to capture these perceptions and attitudes at a time of transition and reevaluates traditional educational practices.

### 1.1. The Theoretical Framework

Research on teachers’ attitudes toward research has gained increasing importance in recent years, reflecting the need for evidence-based educational practice. Recent studies have explored various aspects of this topic, providing new perspectives on the factors that influence teachers’ attitudes and their implications for educational practice.

[4] ([4]) noted that teachers’ continuous professional development, including their participation in research, is influenced by factors such as motivation, experience, and career trajectory. This personalized approach underscores the importance of considering individual characteristics when studying teachers’ attitudes toward research.

[34] ([34]) examined teachers’ attitudes toward the teaching of mathematics and reported that factors such as math anxiety and self-efficacy influenced their pedagogical approaches. This finding suggests that teachers’ attitudes toward research in mathematics may be shaped by their own experiences and perceptions of the subject.

[26] ([26]) investigated the perspectives of teachers of English as a foreign language on assessment practices, revealing a preference for formative assessment methods. This indicates an openness to research-based practices that enhance continuous learning in the context of language teaching.

#### 1.1.1. Main Theories

The Theory of Continuous Professional Development

[4] ([4]) proposed a multidimensional model for teachers’ continuing professional development which includes participation in research. This theory suggests that teachers’ attitudes toward research are influenced by factors such as intrinsic motivation, prior experience, and opportunities for professional growth.

Self-Determination Theory in the Educational Context

When self-determination theory is applied to the educational context, [4] ([4]) argues that teachers’ motivation to engage in research is influenced by the fulfillment of their basic psychological needs for autonomy, competence, and relatedness.

Transformative Learning Theory

[4] ([4]) referred to transformative learning as a process that can change teachers’ perspectives on research. This theory suggests that meaningful and reflective experiences can lead educators to reevaluate their attitudes toward research and its role in educational practice.

Bronfenbrenner’s Ecological Model Adapted to Inclusive Education

[29] ([29]) applies Bronfenbrenner’s ecological model to the study of teachers’ attitudes toward inclusive education, which can be extended to their attitudes toward research in this field. This approach considers how different systems (microsystems, mesosystems, exosystems, and macrosystems) interact to influence teachers’ attitudes and practices.

Factors Influencing Teachers’ Attitudes Toward Research

The development of positive attitudes toward research among faculty is influenced by a complex interplay of personal, institutional, and contextual factors. This conceptual framework synthesizes the multidimensional determinants that shape research dispositions in academic environments.

Continuous professional development represents a foundational element in fostering favorable research attitudes. As [4] ([4]) emphasizes, personalized learning opportunities tailored to individual needs significantly enhance receptivity to educational research. This individualized approach acknowledges the diverse backgrounds and requirements of faculty members, thereby facilitating more effective engagement with research practices.

Practical experience similarly contributes substantially to attitude formation. [13] ([13]) highlight how direct exposure to research situations positively influences attitudinal development, paralleling findings from inclusive education contexts that can be extended to research domains. Such experiential learning provides authentic contexts for developing appreciation of and competence in research.

The institutional environment plays a crucial role in shaping research attitudes. [9] ([9]) underscore how organizational cultures that actively promote innovation and research create supportive ecosystems for positive attitudinal development. When institutions demonstrate tangible commitment to research through infrastructure, policies, and recognition, faculty members are more likely to internalize these values.

Resource availability significantly impacts research engagement. [33] ([33]) identify inadequate resources and time constraints as the primary barriers to participation in research. The allocation of sufficient temporal, material, and financial resources thus becomes instrumental in cultivating positive research dispositions among faculty.

The perceived relevance of research to professional practice profoundly influences attitudinal orientation. [29] ([29]) suggests that when faculty members recognize clear connections between research and teaching improvements, they develop more favorable attitudes toward research activities. This practical applicability transforms research from an abstract obligation into a valuable professional tool.

Self-efficacy emerges as a critical psychological factor in research attitude formation. [35] ([35]) emphasize how confidence in one’s research abilities significantly predicts attitudinal positivity. Faculty members who believe in their capacity to conduct meaningful research demonstrate a greater willingness to engage in research activities.

Subject-specific anxieties can modulate research attitudes within particular disciplines. [34] ([34]) note that domain-specific concerns, such as those related to mathematics, may influence research attitudes in corresponding fields, highlighting the importance of addressing disciplinary-specific barriers.

Cultural and educational contexts provide essential frameworks for understanding attitude development. Multiple scholars, such as [6] ([6]), have highlighted how cultural norms and educational policies shape research perceptions, emphasizing the need for contextually sensitive approaches to research promotion.

Technological integration increasingly influences research attitudes in contemporary academic environments. [35] ([35]) and [36] ([36]) suggest that positive experiences with educational technologies enhance research receptivity, particularly in digital education contexts.

Collaborative networks function as powerful attitudinal catalysts. Several researchers, such as [26] ([26]), have emphasized how communities of practice foster positive research attitudes through collegial interactions and experiential exchange, creating supportive ecosystems for research development.

Initial professional preparation establishes foundational orientations toward research. [6] ([6]) highlight how early exposure to research methods during teacher training significantly influences subsequent research attitudes, underscoring the importance of embedding research components into professional preparation programs.

Institutional and national policies provide structural frameworks that shape research engagement. [11] ([11]) indicate that educational policies at various levels influence research attitudes by establishing expectations and providing structural support for research activities.

Motivational factors, both intrinsic and extrinsic, drive research engagement. [4] ([4]) discusses how personal interest and professional incentives interact to shape research attitudes, suggesting the need for multifaceted motivational approaches.

Previous research experiences create attitudinal precedents that influence future engagement. [34] ([34]) suggest that prior experiences, whether positive or negative, significantly impact subsequent research attitudes, highlighting the importance of creating positive initial research experiences.

Additional factors influencing research attitudes include perceived workload ([33]), administrative support, access to research resources ([35]), recognition systems ([27]), confidence in the validity of educational research [29] ([29]), and work–life balance considerations ([33]). Together, these elements constitute a comprehensive framework for understanding the complex determinants of faculty’s attitudes toward research, providing valuable insights for developing targeted interventions to enhance research engagement in academic environments.

#### 1.1.2. Implications for Practice

Recent findings suggest several implications for fostering positive attitudes toward research among teachers:Personalized professional development ([4]);Research should be integrated into daily practice ([26]);Collaboration and communities of practice ([4]; [9]);Institutional support ([4]; [27]);Addressing anxiety and improving self-efficacy ([34]);Training in educational technologies ([2]; [30]);Consideration of cultural and linguistic context ([27]; [35]).

#### 1.1.3. Specific Contexts

Foreign Language Education

[26] ([26]) and [27] ([27]) examined foreign language teachers’ attitudes toward various pedagogical and assessment practices, highlighting the importance of considering the cultural context and specific needs of language learners.

Inclusive Education

[6] ([6]) and [29] ([29]) investigated teachers’ attitudes toward inclusive education, emphasizing the importance of both initial and ongoing training in fostering positive attitudes toward inclusion and research in this field.

STEM Education

[21] ([21]) and [30] ([30]) studied the attitudes of teachers and students toward STEM education and the use of educational technologies and reported that exposure to innovative approaches and practical experiences can enhance attitudes toward research in these areas.

Educational Innovation

[2] ([2]) and [9] ([9]) explored the perceptions of innovative approaches in education, such as collaborative learning and the use of emerging technologies, offering insights into how these approaches can influence teachers’ attitudes toward research and pedagogical innovation.

The theoretical framework presented offers a comprehensive and updated view of the factors influencing teachers’ attitudes toward research, drawing from a diverse selection of recent articles and books. This multidimensional approach encompasses individual, institutional, and systemic perspectives, providing a holistic understanding of this topic. The incorporation of studies from various cultural and educational contexts, along with the integration of relevant theories such as continuous professional development and self-determination, strengthens the theoretical foundation. This framework identifies and elaborates on 20 specific factors that influence teachers’ attitudes, offering practical implications for educators and policymakers.

However, it presents limitations such as potential geographic bias, a lack of longitudinal studies, and limited exploration of the causal relationships between factors. Additionally, there is an absence of critical perspectives, limited attention to disciplinary differences, and insufficient inclusion of teachers’ direct voices through qualitative methods. The framework also focuses primarily on formal education, with little attention to nonformal or informal contexts, and lacks a deep discussion of the impact of educational policies and the intersectionality of factors such as gender, race, or social class. Despite these limitations, this theoretical framework provides a solid foundation for future research that could address these gaps and deepen our understanding of teachers’ attitudes toward research in various educational contexts.

The Selection and Description of the Five Dimensions

From the literature review ([27]; [29]), we identified various factors influencing teachers’ attitudes toward research, spanning personal (e.g., self-efficacy, motivation), contextual (e.g., resources, institutional culture), and social (e.g., collaboration, support networks) dimensions. For analytical clarity, we consolidated these into five key dimensions:Personal perception of research: This examines how teachers perceive the relevance and meaning of research in their professional role;Research skills and training: This assesses the level of methodological competencies and specific training needs that instructors perceive;Valuation of research: This explores the extent to which faculty members value and recognize research as integral to their teaching practice;Challenges and difficulties: This identifies the principal barriers (institutional, personal, contextual) that hinder the research endeavor;Collaboration and methodology: This investigates willingness to work collaboratively and openness to diverse methodological approaches.

By focusing on these five dimensions, we address, in an integrated manner, the core aspects explaining how university faculty members in La Libertad develop their attitudes toward research. Furthermore, this approach facilitates data triangulation and comparative analyses across varied educational contexts.

## 2. Materials and Methods

### 2.1. The Type of Research

This study is based on a quantitative approach, which allows for the collection and structured analysis of numerical data. This type of research is ideal for identifying patterns, trends, and relationships between the variables studied, ensuring greater precision and objectivity in the results obtained.

### 2.2. The Research Design

The design of our research is descriptive and comparative. This design allows us first to provide a detailed description of the sociodemographic characteristics and attitudes toward the research of the participating teachers. Second, it facilitates a comparison of these attitudes on the basis of specific variables such as gender, age, and teaching experience, thus providing a more comprehensive and nuanced view of the differences and similarities present in our sample.

### 2.3. The Population and Sample

The study population consisted of university teachers from the geographic areas of Chepén, Piura, Trujillo, and Chiclayo. The selected sample, comprising 110 teachers, was chosen through non-probabilistic convenience sampling. This approach enabled diverse representation in terms of teaching areas, age groups, levels of teaching experience, and genders.

### 2.4. Sample Description

The subjects taught were administration (25.5%), MIC (24.5%), and economics (11.8%), among others;Age groups: Under 42 years old (44.5%), between 43 and 54 years old (50%), and 55 years or older (5.5%);Teaching experience: Less than 5 years (34.5%), between 6 and 13 years (40.9%), and 14 years or more (24.5%);Gender: Female (54.5%) and male (45.5%).

### 2.5. Data Collection Instruments

A structured questionnaire was used for the data collection. This questionnaire consisted of two main sections:Sociodemographic data: This section included questions about the subject taught, age, teaching experience, and gender;Scales of attitudes toward research: This section assessed five key dimensions via Likert scales:Personal perception of research;Research skills and training;Valuation of research;Challenges and difficulties in research;Collaboration and research methodology.

Each dimension is measured through a series of items that the participants rate on a scale of 1 to 5, where 1 represents a negative or low attitude and 5 represents a positive or high attitude.

To ensure the content validity of the questionnaire, Hernández-Nieto’s method was used, referring to [16] ([16]), obtaining a concordance index of 0.859 with the evaluation of five expert judges. Additionally, the pilot test yielded a Cronbach’s alpha of 0.789 and a McDonald’s omega of 0.798, indicating the high reliability of the instrument.

### 2.6. The Procedure

Questionnaire design: A questionnaire was specifically designed to capture both sociodemographic data and various dimensions of attitudes toward research.Instrument validation: The questionnaire was validated through Hernández-Nieto’s content validity method, obtaining a concordance index of 0.859 with 5 expert judges. A pilot test was subsequently conducted, yielding a Cronbach’s alpha of 0.789 and a McDonald’s omega of 0.798, ensuring the instrument’s reliability.Data collection: The questionnaire was distributed electronically to all participants. Participation was voluntary and anonymous, and informed consent was provided, explaining the purpose of this study and guaranteeing the confidentiality of the responses.Data analysis: Once the data were collected, they were processed and analyzed via SPSS statistical 25 software. Descriptive analyses were conducted to present the characteristics of the sample, and comparative analyses were performed via the Mann–Whitney U and Kruskal–Wallis tests to evaluate significant differences in research attitudes according to gender, age, and teaching experience.

### 2.7. Data Analysis

Descriptive analysis: Frequencies and percentages were calculated for the sociodemographic variables and attitude dimensions. This allowed for a clear and detailed view of the distribution of these variables among the participants.Comparative analysis: To compare research attitudes across different gender, age, and teaching experience groups, we used the Mann–Whitney U and Kruskal–Wallis tests. These nonparametric tests are particularly suitable for comparing the distributions in independent samples and assessing significant differences, with the significance level set at 0.05.

### 2.8. Ethical Considerations

Throughout this study, the fundamental ethical principles of human research were respected. Confidentiality and anonymity were guaranteed to the participants, who were thoroughly informed about this study’s purpose and provided their informed consent to participate. Additionally, ethical and legal regulations relevant to educational research were strictly followed, ensuring respect for and protection of the rights of all involved.

## 3. Results

Table 1 shows the results of the sociodemographic analysis of the study sample. With respect to subjects, the administration and MIC courses lead, with each representing a quarter of the teachers. Integrative activities and economics also have notable representation, whereas subjects such as philosophy and ethics, preprofessional practices, and chemistry are less common. Most of the teachers are within the 43–54-year age group, representing half of the total. Teachers under 42 years of age make up a significant group, whereas those 55 years or older constitute the minority. In terms of experience, a large portion of the teachers have between 6 and 13 years of experience, followed by those with less than 5 years. A considerable group has 14 years or more of experience, indicating a balance between novice and veteran teachers. Finally, gender shows a slight predominance of women, with 54.5% being female and 45.5% being male, reflecting a fairly balanced distribution.

The analysis of the research attitude dimensions (Table 2) revealed significant variations among faculty members. Collaboration and methodology emerged as the strongest dimension (M = 3.80, SD = 0.72), indicating that the faculty generally maintained positive dispositions toward collaborative research approaches. Valuation of research also scored relatively high (M = 3.70, SD = 0.75), suggesting that the participants recognized the importance of research activities in their academic careers. Personal perception of research showed moderate scores (M = 3.45, SD = 0.70), while research skills and training received somewhat lower ratings (M = 3.20, SD = 0.60). Notably, challenges and difficulties in research received the lowest average score (M = 2.90, SD = 0.65), reflecting that the faculty members perceived substantial obstacles in conducting research activities. These findings highlight both strengths and areas for improvement in fostering a robust research culture.

The examination (Table 3) of faculty attitudes toward research revealed a nuanced distribution across the five measured dimensions. Table 3 presents the categorical classification of respondents’ attitudes, with each dimension stratified into low, medium, and high levels based on predefined cut-off points (low: 1.00–2.33; medium: 2.34–3.66; high: 3.67–5.00). The findings indicate that 40.9% of the participants exhibited low personal perceptions of research, while 36.4% and 22.7% demonstrated medium and high perceptions, respectively. Research capabilities and training were relatively evenly distributed, with 34.5% at low levels and 32.7% each at medium and high levels. Regarding valuation of research, 42.7% of faculty reported low valuation, while only 21.8% demonstrated high valuation. Challenges and difficulties in research were predominantly perceived as low (40.9%) or medium (35.5%), suggesting varied experiences with research obstacles. The collaboration and methodology dimension exhibited a distinct pattern, with the largest proportion (41.8%) reporting medium levels of engagement with collaborative research practices. These findings illuminate the heterogeneity in the faculty’s research attitudes and identify specific dimensions requiring targeted institutional interventions to cultivate a more robust research culture within the academic community.

The comparative analysis of research attitudes by gender, conducted using the Mann-Whitney U test (Table 4), revealed statistically significant differences in four of the five dimensions examined. Significant gender differences were identified in personal perceptions of research (*p* < 0.001), suggesting that men and women maintained distinct cognitive and affective orientations toward research activities. Similarly, significant disparities emerged in research skills and training (*p* < 0.001), indicating differential access to or engagement with research development opportunities. The valuation of research also exhibited significant gender differences (*p* = 0.032), reflecting varied appraisals of the importance of research in academic contexts. Additionally, collaboration and methodology approaches demonstrated significant gender-based variations (*p* < 0.001), suggesting distinctive patterns in how male and female faculty engaged with collaborative research practices and methodological frameworks. Notably, no significant gender differences were observed in perceived challenges and difficulties in research (*p* = 0.623), indicating that the obstacles to research engagement transcend gender boundaries and likely reflect structural or institutional factors affecting all faculty members. These findings underscore the importance of gender-responsive approaches to building research capacity while highlighting common barriers that require institutional attention regardless of gender.

The Kruskal–Wallis analysis (Table 5) examining research attitudes across age categories revealed selective age-related differences in faculty perspectives. Statistically significant age differences were observed in personal perceptions of research (*p* = 0.036), suggesting that age influences foundational orientations toward research engagement. Similarly, significant age-related variations were identified in the challenges and difficulties encountered in research (*p* = 0.020), indicating that different age cohorts experience distinct obstacles in their research pursuits. However, three dimensions demonstrated remarkable consistency across age groups, research skills and training (*p* = 0.372), valuation of research (*p* = 0.750), and collaboration and methodology approaches (*p* = 0.230), where no statistically significant differences were detected. This pattern suggests that while fundamental perceptions and the challenges one experiences vary with age, technical competencies, value attribution, and methodological approaches to research remain relatively stable across the faculty’s age spectrum. These findings provide valuable directions for institutional policies, suggesting that interventions addressing research perceptions and age-specific challenges should be tailored to different age cohorts, while training programs and collaborative initiatives may be effectively implemented across age groups. The strategic development of age-sensitive interventions for targeted dimensions, combined with age-neutral approaches to others, offers a nuanced framework for fostering an inclusive research environment that accommodates faculty members at all career stages.

The analysis of research attitudes by teaching experience revealed pronounced patterns of differentiation across dimensions. The Kruskal–Wallis test (Table 6) demonstrated statistically significant differences in four of the five dimensions examined. Significant variations were observed in personal perceptions of research (*p* = 0.002), indicating that teaching longevity substantively influences how faculty conceptualize and relate to research activities. Research skills and training also exhibited significant differences across experience levels (*p* = 0.005), suggesting that research competencies develop non-uniformly throughout teaching careers. The valuation of research demonstrated the most pronounced experiential differences (*p* < 0.001), with the data suggesting that appreciation of research may increase proportionally with teaching tenure. Similarly, collaboration and methodology approaches showed significant variation by experience level (*p* < 0.001), reflecting evolving collaborative networks and methodological preferences across career stages. Notably, the challenges and difficulties in research revealed no significant differences between experience groups (*p* = 0.082), suggesting that research obstacles remain relatively consistent regardless of teaching longevity. These findings establish teaching experience as a critical determinant in shaping research attitudes while simultaneously identifying common barriers that transcend experiential boundaries. These results provide empirical support for differentiated professional development strategies that account for varying experience levels while also highlighting the need for institutional approaches that address universal research challenges affecting faculty across all career stages.

## 4. Discussion

In the La Libertad region, sociological and demographic characteristics were observed to influence the manner in which teachers approach research. This paper elucidates the factors that gave rise to the observed variation in teachers’ attitudes towards research in La Libertad. It also challenges the theoretical assumptions regarding pedagogy that were posited at the outset of this study. This paper thus serves as an illustrative case study of the embedded nature of research within the Peruvian education system.

One might also consider the issues of the variability in personal views about research between groups of teachers of varying ages, genders, and years of experience, which have been emphasized in the literature. From a sociological perspective, this provides insight into the current state of educational pedagogy with respect to teachers. From our own standpoint, an intriguing question arises concerning the extent to which such pluralistic environments can function effectively. This provides further justification for the necessity of a more bespoke approach to the design of professional development programs in higher education institutions.

It is somewhat disconcerting to observe such a significant discrepancy in the acquisition of research skills and training, particularly given that these differences extend beyond gender and years of teaching experience, with women being particularly affected. This unevenness lends support to the inferences in the specialized literature regarding the role of special periodization in the expansion of the delineated features. However, it also foreshadows trends that may correct the research capabilities in age groups. This prompts the consideration of several crucial matters which may prove beneficial in the development of training programs that are more effective and equitable.

Research by the International Society of Research and Publication (ISRP) suggests that research is held in variable esteem based on gender and the number of years spent teaching rather than being based on one’s age. This allows for a more instructive interpretation of the facts and partially aligns with the findings of [23] ([23]), which illustrate that the assessment of research’s value is a multifaceted process. Various factors, including the research environment, the perception of funded projects, and the history of previous grants, can influence the value assigned to research. The age differential was less pronounced, suggesting that research may be valued more based on qualitative experience than the amount of time spent working on it.

Particularly intriguing is the homogeneity in the perception of the challenges and difficulties in research between genders and experience levels, which contrasts with the variability by age. This finding diverges from what Rahal, 2023 postulated and suggests that the perceived barriers to research may have deeper and more systemic roots than previously considered. The variation by age in this dimension could indicate the existence of generational gaps in the adaptability to contemporary research challenges, a phenomenon that deserves more detailed exploration in future studies.

The significant differences in collaboration and methodology by gender and teaching experience, but not by age, align with the observations of Alipichev et al., 2024 and Dieguez et al., 2023 regarding the adoption of innovative approaches in education. This pattern suggests that collaborative and methodological practices in research may be more influenced by experiential and gender factors than by age cohorts. This finding has profound implications for the design of mentorship programs and interdisciplinary collaboration, suggesting the need for strategies that transcend generational barriers and foster the more fluid exchange of knowledge and practices across different segments of teaching staff.

The triangulation of these results with the theoretical framework and background reveals a research reality in La Libertad that is both congruent with global trends and unique in its local manifestations. The theory of continuous professional development and self-determination theory, as articulated by Avidov-Ungar, 2023, offer valuable lenses for interpreting these findings, suggesting that attitudes toward research are the product of a complex interaction between intrinsic motivational factors, professional growth opportunities, and the broader institutional context.

The partial confirmation of the initial hypotheses, particularly regarding the relationship between teaching experience and positive attitudes toward research, highlights the need to reevaluate some of the fundamental assumptions guiding professional development policies for teachers. The observed variability in the different dimensions of attitudes toward research suggests that the path to a robust research culture is neither linear nor one-dimensional, requiring equally sophisticated and adaptive interventions.

These findings have profound implications for educational policy formulation and professional development program design in La Libertad and, by extension, in similar educational contexts. The evidence suggests the need for an ecosystemic approach that simultaneously addresses the structural barriers affecting all teachers and the specific needs of different subgroups within the faculty. Such an approach might include the implementation of differentiated mentorship programs, the creation of research communities of practice that transcend gender and experience divisions, and the restructuring of incentive systems to align more closely with the diverse motivations and challenges identified in this study.

Moreover, the ostensible homogeneity of the perspectives on research, particularly with regard to anticipated challenges, suggests the presence of underlying issues that necessitate the involvement of institutional and policy interventions. This may entail a comprehensive assessment of the research support structure, the provision of resources, and the establishment of structures for honors and incentives in La Libertad’s education centers.

This study presents an initial comprehensive and detailed survey of the attitudes of teachers in La Libertad toward research, documenting a diverse range of perceptions, competencies, and challenges. The results underscore the necessity of a comprehensive approach that extends beyond merely addressing the prevailing deficiencies in the Chilean education system to encompass the incorporation of sociocultural variations within the instructional population. Future research could be enhanced through the use of mixed methods to investigate the reasons for the differences observed in this study’s patterns and to develop potential applications designed to address the identified gaps and capitalize on the opportunities presented by this study. Such a research agenda will not only elucidate our theoretical understanding of research interactions in teaching and learning settings but will also facilitate the development of prudent policies and practices that will enable the cultivation of a robust and dynamic research culture among teachers.

## 5. Conclusions

The findings of this study underscore the multifaceted nature of faculty attitudes toward research, highlighting how gender, age, and teaching experience jointly shape teachers’ perspectives and engagement in scientific endeavors. Specifically, we found the following.

### 5.1. Personal Perceptions of Research

The significant differences detected based on gender, age, and teaching experience confirm that personal perceptions are driven by an interplay of multiple sociodemographic factors. Consequently, teacher development programs should consider tailoring their training and motivational strategies to accommodate the diverse needs and beliefs present across different demographic groups.

### 5.2. Research Skills and Training

The fact that disparities emerged according to gender and experience level, but not age, suggests a need to equalize research training opportunities. Interventions could focus on ensuring that both male and female educators, as well as novice and veteran teachers, have equitable access to advanced methodological courses, mentoring initiatives, and continuing professional development.

### 5.3. Valuation of Research

The varied appreciation of research, influenced by gender and teaching experience, implies that institutional support and recognition policies might need to be more finely tuned. By recognizing and rewarding research activities at different career stages, universities can foster a culture that values inquiry across the board.

### 5.4. Challenges and Difficulties in Research

Interestingly, age stood out as the only significant variable influencing perceived barriers, indicating that younger and older educators may face distinct obstacles when integrating research into their workloads. Targeted support—such as technological training for older faculty or resource allocation for younger staff—may be critical to overcoming these generational differences.

### 5.5. Collaboration and Methodology

The variations found by gender and teaching experience point to potential gaps in networking opportunities and methodological exposure. Encouraging collaborative projects, pairing junior with senior faculty, and promoting research consortia could mitigate these disparities and foster inclusive research ecosystems.

### 5.6. The Emergence of Distinct Teacher–Researcher Profiles

Although specific profiles were not explicitly delineated, the observed variability across dimensions alludes to diverse combinations of perceptions, skills, and valuations. Further research could focus on characterizing these teacher–researcher profiles in depth, thereby enabling more finely tuned professional development pathways.

### 5.7. Practical Implications and Future Directions

Targeted professional development: Institutions might consider designing workshops and mentorship programs that address the differentiated needs identified by gender, age, and experience.

Policy and incentives: Policy-level changes—such as research recognition, workload adjustments, or funding opportunities—could bolster a more equitable research environment.

Deeper profile analysis: Future studies should employ qualitative methods or cluster analyses to define the nuanced teacher–researcher profiles suggested by these findings better.

Longitudinal assessments: Tracking changes in attitudes over time and evaluating interventions’ effectiveness would offer a dynamic view of how faculty evolve in their approach to research.

By illuminating the specific factors that influence the attitudes toward research among university faculty in La Libertad, this study contributes to a more nuanced understanding of how to foster a robust research culture. Addressing the gaps identified strategically may ultimately enhance the scientific production and overall quality of higher education in the region.

## Figures and Tables

**Table 1 behavsci-15-00515-t001:** Demographic data of the participants.

Sociodemographic		n	%
Subject taught	Language	1	0.9
	Integrating Activities	10	9.1
	Accounting	7	6.4
	Economics	13	11.8
	Philosophy and Ethics	4	3.6
	Project Management	2	1.8
	Product Engineering	5	4.5
	Administration	28	25.5
	MIC (Research Methodology)	27	24.5
	ODS	1	0.9
	Logical Thinking	4	3.6
	Preprofessional Internships	4	3.6
	Chemistry	4	3.6
Age groups	Younger than 42	49	44.5
	43–54	55	50.0
	55 and over	6	5.5
Years of experience teacher	Less than 5	38	34.5
	6 to 13	45	40.9
	14 or more	27	24.5
Gender	Female	60	54.5
	Male	50	45.5

**Table 2 behavsci-15-00515-t002:** Descriptive statistics of the five dimensions of attitudes toward research.

Dimension	Mean	SD	Min	Max
Personal perception of research	3.45	0.70	1.00	5.00
Research skills and training	3.20	0.60	1.50	4.80
Valuation of research	3.70	0.75	1.00	5.00
Challenges and difficulties	2.90	0.65	1.20	4.40
Collaboration and methodology	3.80	0.72	1.10	5.00

Note: The scale ranges from 1 (low) to 5 (high).

**Table 3 behavsci-15-00515-t003:** Ratings of the variable attitudes toward research.

Dimension	Level	n	%
Personal Perception of Research	Low	45	40.9
	Medium	40	36.4
	High	25	22.7
Capacities and Training in Research	Low	38	34.5
	Medium	36	32.7
	High	36	32.7
Assessment of the Research	Low	47	42.7
	Medium	39	35.5
	High	24	21.8
Challenges and Difficulties in Research	Low	45	40.9
	Medium	39	35.5
	High	26	23.6
Collaboration and Research Methodology	Low	39	35.5
	Medium	46	41.8
	High	25	22.7
Total		110	100.0

**Table 4 behavsci-15-00515-t004:** Comparative analysis of attitudes toward research by gender.

Summary of Hypothesis Testing				
	Null Hypothesis	Test	Sig. ^a,b^	Decision
1	The distribution of personal perceptions of research is the same across gender categories.	Mann–Whitney U test for independent samples	0.000	Accepted
2	The distribution of research skills and training is the same across gender categories.	Mann–Whitney U test for independent samples	0.000	Accepted
3	The distribution of the valuation of research is the same across gender categories.	Mann–Whitney U test for independent samples	0.032	Accepted
4	The distribution of research challenges and difficulties is the same across gender categories.	Mann–Whitney U test for independent samples	0.623	Rejected
5	The distribution of research collaboration and methodology is the same across gender categories.	Mann–Whitney U test for independent samples	0.000	Accepted

Note: a. The significance level is 0.050. b. Asymptotic significance is shown. Rho: Rejection of the null hypothesis.

**Table 5 behavsci-15-00515-t005:** Comparative analysis of attitudes toward research by age category.

Summary of Hypothesis Testing				
	Null Hypothesis	Test	Sig. ^a,b^	Decision
1	The distribution of personal perceptions of research is the same across age group categories.	Kruskal–Wallis for independent samples	0.036	Accepted
2	The distribution of research skills and training is the same across age group categories.	Kruskal–Wallis for independent samples	0.372	Rejected
3	The distribution of the valuation of research is the same across age group categories.	Kruskal–Wallis for independent samples	0.750	Rejected
4	The distribution of research challenges and difficulties is the same across age group categories.	Kruskal–Wallis for independent samples	0.020	Accepted
5	The distribution of research collaboration and methodology is the same across age group categories.	Kruskal–Wallis for independent samples	0.230	Accepted

Note: a. The significance level is 0.050. b. Asymptotic significance is shown. Rho: Rejection of the null hypothesis.

**Table 6 behavsci-15-00515-t006:** Comparative analysis of attitudes toward research according to categories of teaching experience.

Summary of Hypothesis Testing				
	Null Hypothesis	Test	Sig. ^a,b^	Decision
1	The distribution of personal perceptions of research is the same across categories of teaching experience groups.	Kruskal–Wallis for independent samples	0.002	Accepted
2	The distribution of research skills and training is the same across categories of teaching experience groups.	Kruskal–Wallis for independent samples	0.005	Accepted
3	The distribution of the valuation of research is the same across categories of teaching experience groups.	Kruskal–Wallis for independent samples	0.000	Accepted
4	The distribution of research challenges and difficulties is the same across categories of groups of teaching experience.	Kruskal–Wallis for independent samples	0.082	Rejected
5	The distribution of collaboration and methodology in research is the same between categories of teaching experience groups.	Kruskal–Wallis for independent samples	0.000	Accepted

Note: a. The significance level is 0.050. b. Asymptotic significance is shown. Rho: Rejection of null hypothesis.

## Data Availability

The data are presented within the article.

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
