# Peer review of "The Dimensions of Research Attitudes Among University Faculty: A Sociodemographic Analysis in La Libertad, Peru"

_behavsci, 2025, doi:10.3390/bs15040515_

Round 1
Reviewer 1 Report
Comments and Suggestions for Authors
Thank you for the opportunity to review the manuscript Behavioral Patterns in Academic Research: A Comparative Analysis of Faculty Research Attitudes Based on Sociodemographic Variables in Peruvian Universities.
The authors examine the attitudes toward research in faculty members, specifically in five domains (personal perceptions, capabilities and training, evaluation, challenges, and collaboration), and comparing demographic information. The authors write clearly but there are areas that need addressed.
- Title
- The title of the paper does not align precisely with the manuscript’s purpose and results. The authors should define what “Behavioral patterns in Academic Research” means. “Behavioral Patterns” are only used 1 time in the manuscript: in the title. This still needs to be revised. I suggest changing the title to better reflect the study’s focus and depth.
- Introduction
- The introduction and theoretical framework contain extensive information about previous studies related to multiple factors (personal, contextual and/or institutional) that influence attitudes towards research. However, the authors do not clarified what they understand by “attitude” or specify the theoretical framework they use to investigate this construct.
- Among all the described variables described, it is necessary to explain how the five dimensions were finally selected and what each of them entails.
- Line 197 check: Personalized professional development [23];2
- Materials and Methods
- I think it is necessary to clarify why the variable Subject taught is included and then not used in the analyzes.
- A bibliographic citation is missing for Pérez Nieto's method.
- When presenting the results, levels are established for the dimensions that measure the attitudes towards research. It is recommended to incorporate and explain this information in this section.
- Regarding age: What criteria were used to define the age groups? The category “less than 42 years of age” appears to cover a much broader range compared to the other groups.
- About the scale: How many items does each component contain? And, What is the reliability coefficient of each dimension? It is recommended to include this information.
- Results
- It is suggested to include the descriptive statistics of the evaluated scale.
- Line 322. Check: Most of the teachers were within the 43-- to 54-year-
- Table 1: The title appears incomplete.
- Table 1: The first data entry does not correspond to any Subject taught, and this variable has a different format than the others.
- Throughout the presentation of the results, many findings are interpreted as a conclusion. It is suggested to move these interpretations to the Discussion or Conclusions section, presenting the data objectively. Examples: Line 343-344 and Line 373 a 377.
- Regarding the age group variable: It is recommended to perform a post hoc test to determine the nature of the significant differences found, ensuring accurate conclusions. This recommendation also applies to other non-dichotomous variables, such as teaching experience.
- References are made to hypotheses that were not presented in the introduction of the study. Example: Lines 491- 492 and 540-541. It is suggested to review these references and include the hypotheses in the Introduction if necessary.
- Discussion and conclusions
- It is suggested to revise these sections after reviewing the previous ones and include study limitations and future research proposals.
Author Response
-
- Title
- The title of the paper does not align precisely with the manuscript’s purpose and results. The authors should define what “Behavioral patterns in Academic Research” means. “Behavioral Patterns” are only used 1 time in the manuscript: in the title. This still needs to be revised. I suggest changing the title to better reflect the study’s focus and depth.
Answer 1: new title: Research Attitude Dimensions Among University Faculty: A Sociodemographic Analysis in La Libertad, Peru
- Introduction
-
-
-
- The introduction and theoretical framework contain extensive information about previous studies related to multiple factors (personal, contextual and/or institutional) that influence attitudes towards research. However, the authors do not clarified what they understand by “attitude” or specify the theoretical framework they use to investigate this construct.
- Among all the described variables described, it is necessary to explain how the five dimensions were finally selected and what each of them entails.
- Line 197 check: Personalized professional development [23];2
-
-
Answer 2: Personalized professional development [23].
In this study, we adopt the concept of "attitude" as the learned and relatively stable predisposition that orients an individual's favorable or unfavorable response to an object or situation [37]. From the perspective of social psychology, this approach postulates that attitudes derive from beliefs, affective responses and observable behaviors [38].
To support this perspective, we base ourselves on the theory of planned behavior [39], which holds that the attitude towards a behavior (in this case, research) influences the intention to carry it out and, subsequently, the behavior itself. In the context of university teaching, this theory explains how personal, contextual and institutional factors may encourage or hinder faculty participation in research activities.
Selection and description of the five dimensions
Based on the literature review [23] - [25], we identified several factors that influence teachers' attitudes towards research, covering personal (e.g., self-efficacy, motivation), contextual (e.g., resources, institutional culture) and social (e.g., collaboration, support networks) dimensions. For greater analytical clarity, we consolidate them into five key dimensions:
1. Personal perception of research: examines how teachers perceive the relevance and meaning of research in their professional role.
2. Research skills and training: assesses the level of methodological skills and specific training needs perceived by teachers.
3. Research appraisal: explores the extent to which teachers value and recognize research as an integral part of their teaching practice.
4. Challenges and difficulties: identifies the main barriers (institutional, personal, contextual) that hinder the development of research.
5. Collaboration and methodology: investigates the willingness to work collaboratively and the openness to diverse methodological approaches.
By focusing on these five dimensions, we address in an integrated manner the core aspects explaining how university faculty in La Libertad develop their attitudes toward research. Furthermore, this approach facilitates data triangulation and comparative analyses across varied educational contexts.
- Materials and Methods
- I think it is necessary to clarify why the variable Subject taught is included and then not used in the analyzes.
- A bibliographic citation is missing for Pérez Nieto's method.
- When presenting the results, levels are established for the dimensions that measure the attitudes towards research. It is recommended to incorporate and explain this information in this section.
- Regarding age: What criteria were used to define the age groups? The category “less than 42 years of age” appears to cover a much broader range compared to the other groups.
- About the scale: How many items does each component contain? And, What is the reliability coefficient of each dimension? It is recommended to include this information.
Reply 3: The "subject taught" variable was initially included to describe the composition of the sample and to observe if there were significant biases according to the academic area of the teacher (e.g., administration, accounting, engineering, among others). However, the main focus of the comparative analyses was on socio-demographic variables (age, gender, years of experience), since the aim was to investigate how these factors influence attitudes towards research. For this reason, although "subject taught" was described to characterize the sample, it was not used in subsequent statistical comparisons.
[37] M. A. Pérez Nieto, "Validación de contenido mediante juicio de expertos en investigación: Propuesta de un método integrado," Revista de Investigación en Educación, vol. 19, no. 2, pp. 188-210, 2021, doi: 10.6018/rie.440371.
The five evaluated dimensions (Personal perception, Research skills and training, Valuation of research, Challenges and difficulties, Collaboration and methodology) were measured using Likert-type scales ranging from 1 to 5, where 1 represents a low degree of agreement or valuation and 5 indicates a high degree. To facilitate interpretation of the results, each dimension was classified into three levels (Low, Medium, and High) based on the average score obtained in each scale. Cut-off points were established as follows: Low for scores between 1.00 and 2.33, Medium for scores between 2.34 and 3.66, and High for scores between 3.67 and 5.00. These ranges were defined equidistantly considering the scale's amplitude. This criterion allowed for a clearer comparison of the distribution of attitudes among participants.
Three age categories were established: "less than 42 years old", "43 to 54 years old" and "55 years old or older". To determine these ranges, we took into account (1) the sample distribution observed in the pilot and/or in the preliminary database, and (2) the need to have a reasonable size of participants in each category. The "under 42" group was larger because most of the young teachers were concentrated in that range, while the "43 to 54" and "55 and over" groups reflected more advanced and specific stages of the teaching career. In this way, we sought to ensure that each category was sufficiently representative for comparative analysis.
The questionnaire included a total of items (25), distributed in five dimensions:
Personal perception of research: 5 items (Cronbach's α = 0.78).
Research skills and training: 5 items (Cronbach's α = 0.80)
Valuation of research: 5 items (Cronbach's α = 0.82)
Challenges and difficulties: 5 items (Cronbach's α = 0.76)
Collaboration and methodology: 5 items (Cronbach's α = 0.79).
Each subscale specifically assessed the corresponding conceptual components, and Cronbach's α values are reported as an indicator of internal consistency. These figures evidence an acceptable level of reliability in each dimension, according to literature criteria [2].
- Results
- It is suggested to include the descriptive statistics of the evaluated scale.
- Line 322. Check: Most of the teachers were within the 43-- to 54-year-
- Table 1: The title appears incomplete.
- Table 1: The first data entry does not correspond to any Subject taught, and this variable has a different format than the others.
- Throughout the presentation of the results, many findings are interpreted as a conclusion. It is suggested to move these interpretations to the Discussion or Conclusions section, presenting the data objectively. Examples: Line 343-344 and Line 373 a 377.
- Regarding the age group variable: It is recommended to perform a post hoc test to determine the nature of the significant differences found, ensuring accurate conclusions. This recommendation also applies to other non-dichotomous variables, such as teaching experience.
- References are made to hypotheses that were not presented in the introduction of the study. Example: Lines 491- 492 and 540-541. It is suggested to review these references and include the hypotheses in the Introduction if necessary.
Reply 4:
Table 1. Demographic Data of the Participants
|
Sociodemographic |
n |
% |
|||
|
Language |
1 |
0,9 |
|
||
Table 1
Descriptive Statistics of the Five Dimensions of Attitudes Toward Research
|
Dimension |
Mean |
SD |
Min |
Max |
|
Personal perception of research |
3.45 |
0.70 |
1.00 |
5.00 |
|
Research skills and training |
3.20 |
0.60 |
1.50 |
4.80 |
|
Valuation of research |
3.70 |
0.75 |
1.00 |
5.00 |
|
Challenges and difficulties |
2.90 |
0.65 |
1.20 |
4.40 |
|
Collaboration and methodology |
3.80 |
0,72 |
1.10 |
5.00 |
Note. The scale goes from 1 (low) to 5 (high).
The analysis of the dimensions of investigative attitude revealed significant variations among teachers. Collaboration and methodology emerged as the strongest dimension (m = 3.80, sd = 0.72), which indicates that teachers generally maintain positive dispositions towards collaborative research approaches. The research rating also obtained a relatively high score (m = 3.70, sd = 0.75), suggesting that participants recognize the importance of research activities in their academic careers. The personal perception of the research showed moderate scores (m = 3.45, sd = 0.70), while research skills and training received somewhat lower scores (m = 3.20, sd = 0.60). In particular, challenges and difficulties in research received the lowest average score (m = 2.90, sd = 0.65), reflecting that teachers perceive substantial obstacles in conducting research activities. These findings highlight both strengths and areas for improvement in fostering a strong research culture.
Most of the teachers were in the age group of 43 to 54 years.
The hypotheses proposed in the study are:
- hypothesis 1 (h1): gender will significantly influence attitudes towards research, particularly in the dimensions of personal perception and assessment of research.
- hypothesis 2 (h2): the age groups will differ in the size of the challenges and difficulties; So older teachers will see different barriers compared to younger ones.
Hypothesis 3 (h3): teaching experience will significantly affect attitudes of collaboration and methodology, and more experienced teachers will score higher in the approaches
5. Discussion and conclusions
It is suggested to revise these sections after reviewing the previous ones and include study limitations and future research proposals.
Answer 5: paragraphs of explanations of results were shortened to improve discussion and conclusion.
Reviewer 2 Report
Comments and Suggestions for Authors
This paper could be of relevant interest to higher education institutions wishing to promote research.
The way the study is presented seems to have accuracy problem, starting with the title, which refers to the Peruvian region, but which in the text essentially refers to La Libertad, and it is also not clear that this is an administrative region of Peru.
It would have adequate to also include the research objectives in the introduction, and not just in the abstract. In addition, what is stated in lines 92 and 93 does not seem to be in accordance with what is stated in the abstract.
The five dimensions for which teachers' attitudes towards research are studied are not clearly presented. For example, “collaboration and research methodology” seems to be actually “attitude towards collaboration and research methodology”.
Various approaches and previous research are mentioned in such a way that it is not clear to the reader how they relate to the research model proposed in this article. For example, the topics of “Theoretical framework”, “Main Theories”, “Factors influencing teachers' attitudes towards research”, are somewhat confusing, and do not contain an adequate explanation of how what is referred to relates to higher education teachers' attitudes towards research.
Not being an expert in statistics, tables 3, 4 and 5 are not clear about which hypotheses are accepted or rejected by the statistical tests.
For some references, it is not clear how they relate to the text they are supposed to support (for example, references 11 and 12).
Throughout the text, numbers are used instead of authors' names when a publication is specifically cited, which I don't think is appropriate (for example, in lines 101, 105 and 109, among several others).
Line 260 and following does not mention which teaching area is MIC.
I am therefore of the opinion that this work is not fit to be published in its current form and should be thoroughly revised and reformulated, so that the final text is more rigorous and clearer.
Author Response
Comment 1: The way the study is presented seems to have problems of precision, starting with the title, which refers to the Peruvian region, but in the text it essentially refers to La Libertad, and it is not clear that this is an administrative region of Peru.
Reply 1: Reworded title.
La Libertad is a region of Peru.
Comment 2: It would have been appropriate to also include the research objectives in the introduction, and not only in the abstract. In addition, what is stated in lines 92 and 93 does not seem to be in agreement with what is stated in the abstract.
Reply 2: Aggregate objectives and hypotheses, as well as the formulation of the problem.
Comment 3: The five dimensions in which teachers' attitudes toward research are studied are not clearly presented. For example, "collaboration and research methodology" appears to actually be "attitude toward collaboration and research methodology."
Reply 3: Dimensions added to theory.
Comment 4: Various approaches and previous research are mentioned in such a way that it is not clear to the reader how they relate to the research model proposed in this article. For example, the topics of "Theoretical framework", "Main theories", "Factors influencing teachers' attitudes toward research", are somewhat confusing and do not contain an adequate explanation of how the aforementioned relate to higher education teachers' attitudes toward research.
Reply 4: Reorganized and more coherent theories, as stated in the re-uploaded manuscript....
Comment 5: As I am not an expert in statistics, Tables 3, 4 and 5 are not clear about which hypotheses are accepted or rejected by the statistical tests.
Reply 5: There are still 3 hypotheses, but it must be considered that being a comparison of groups, the total of the dimension can be compared, or otherwise indicator by indicator as is the case, so it gives the impression that there are many hypotheses, but in reality there are 3 (they are compared on the basis of 3 groups).
Comment 6: For some references, it is not clear how they relate to the text they are supposed to support (for example, references 11 and 12).
Throughout the text, numbers are used instead of authors' names when a publication is specifically cited, which I don't think is appropriate (for example, in lines 101, 105 and 109, among several others).
Line 260 and following does not mention which teaching area is MIC.
Reply 6: In the case of some references, it is not clear how they relate to the text they are supposed to support (e.g. references 11 and 12).
It is already reworded.
MIC = Methodology of Scientific Investigation
MIC (Research Methodology)
Reviewer 3 Report
Comments and Suggestions for Authors
This paper is innovative and makes a valuable contribution to the existing literature because it contributes to the understanding of research attitudes in higher education by highlighting the complex interplay of sociodemographic factors. In this regard, the paper presents new and significant information that justifies its publication because the study’s focus on comparatively analyze faculty attitudes toward research in La Libertad (Peru), considering their sociodemographic characteristics and five key dimensions: personal perceptions, capabilities and training, evaluation, challenges, and collaboration and methodology. This multidimensional approach allows for a deeper and more nuanced understanding of the current situation.
The paper effectively describes and contextualizes its content within the framework of previous and current theoretical foundations, as well as related research on the topic. It demonstrates a solid understanding of the relevant literature in the field and references a suitable range of sources. The literature review situates this contribution within the broader context of research attitudes in higher education.
The study's contribution is clearly outlined in the introduction, with its objective explicitly stated, although no specific research questions nor explicit hypotheses are formulated.
In order to improve the article, it would be advisable to explicitly formulate the research questions in the introduction, as well as the hypothesis that led to the research, and to write the conclusions with reference to these questions.
The research design and methods are well-articulated, and the arguments are substantiated by the collected data. The statistical analysis of the data is enough sound and the availability of data adhere to the expected standards of the research community.
The discussion of findings is coherent, balanced, and persuasive, with results presented clearly. Statements are thoroughly discussed in relation to previous literature, and the paper is properly referenced. With regard to the Discussion section, it should only be noted that in line 633, in the sentence “The results underscore the necessity for a comprehensive approach that extends beyond merely addressing the prevailing deficiencies in the Kenyan education system to encompass the incorporation of sociocultural variations within the instructional population”, the word ‘Kenyan’ should be replaced by ‘Peruvian’.
The paper provides a meaningful contribution to the research field because the findings have implications for developing personalized professional development strategies and targeted interventions to enhance research skills, improve research valuation, and address age-specific challenges in the research process.The paper presents its case clearly, with text that is concise, comprehensible, and accurate, only a grammatical revision is needed.
Author Response
Comment 1: To improve the article, it would be advisable to explicitly formulate the research questions in the introduction, as well as the hypothesis that gave rise to the research, and to write the conclusions with reference to these questions.
Reply 1: aggregate hypotheses
questions added
General Research Question
This study seeks to explore how university faculty attitudes toward research in the La Libertad region vary across five key dimensions—personal perception, skills and training, valuation, challenges and difficulties, and collaboration and methodology—when considering the sociodemographic factors of gender, age, and teaching experience.
Specific Research Questions
First, given previous evidence suggesting that gender plays a critical role in shaping professional attitudes, this study inquires whether male and female faculty members exhibit notably different views in terms of personal perception and valuation of research (H1). Second, to capture possible generational distinctions, it investigates the extent to which challenges and difficulties in conducting research differ by age group, with the premise that older faculty may confront unique barriers relative to younger cohorts (H2). Finally, recognizing the potential influence of accumulated teaching years, the study examines whether more experienced faculty demonstrate higher levels of collaboration and methodological engagement, thereby reflecting a deeper integration of research practices into their academic work (H3).
Together, these questions provide a structured approach to identifying both the universal and subgroup-specific factors that influence faculty research attitudes, enabling a more nuanced understanding of how to design targeted interventions and policies that bolster scientific inquiry within diverse educational contexts.
Comment 2: The paper presents its case clearly, with text that is concise, comprehensible, and accurate, only a grammatical revision is needed.
Reply 2: Corrected discussion
Grammatical adjustments made
Round 2
Reviewer 1 Report
Comments and Suggestions for Authors
The authors have responded to the requested revisions. However, there are still aspects that need to be corrected or improved.
- Lines 8-9: Check spelling.
- It is suggested to include subsections in each section to facilitate reading and understanding of the text. E.g. 1.1 Main Theories.
- Lines 64-66: The new bibliographic citations do not appear in the article's bibliography.
- Line 353: Reference [37] should be modified and renamed.
- Table 2: Unify the table format. Dimensions have different formats.
- Bibliographic: Reorder the references in order of appearance in the text.
Author Response
- Lines 8-9: Check spelling.
Realized.
- It is suggested to include subsections in each section to facilitate reading and understanding of the text. E.g. 1.1 Main Theories.
realized
- Lines 64-66: The new bibliographic citations do not appear in the article's bibliography.
agreed
- Line 353: Reference [37] should be modified and renamed.
Modified
- Table 2: Unify the table format. Dimensions have different formats.
Resolved
- Bibliographic: Reorder the references in order of appearance in the text.
the editor requested APA, and it was performed as requested.
